# Bioinspired Cyclic Dipeptide Functionalized Nanofibers for Thermal Sensing and Energy Harvesting

**DOI:** 10.3390/ma16062477

**Published:** 2023-03-21

**Authors:** Daniela Santos, Rosa M. F. Baptista, Adelino Handa, Bernardo Almeida, Pedro V. Rodrigues, Ana R. Torres, Ana Machado, Michael Belsley, Etelvina de Matos Gomes

**Affiliations:** 1Laboratory for materials and Emergent Technologies (LAPMET), Centre of Physics of Minho and Porto Universities (CF-UM-UP), University of Minho, Campus de Gualtar, 4710-057 Braga, Portugal; 2Institute for Polymers and Composites, University of Minho, Campus de Azurém, 4800-058 Guimaraes, Portugal

**Keywords:** cyclic dipeptides, biopolymers, electrospinning, photoluminescence, nanofibers, energy harvesting

## Abstract

Nanostructured dipeptide self-assemblies exhibiting quantum confinement are of great interest due to their potential applications in the field of materials science as optoelectronic materials for energy harvesting devices. Cyclic dipeptides are an emerging outstanding group of ring-shaped dipeptides, which, because of multiple interactions, self-assemble in supramolecular structures with different morphologies showing quantum confinement and photoluminescence. Chiral cyclic dipeptides may also display piezoelectricity and pyroelectricity properties with potential applications in new sources of nano energy. Among those, aromatic cyclo-dipeptides containing the amino acid tryptophan are wide-band gap semiconductors displaying the high mechanical rigidity, photoluminescence and piezoelectric properties to be used in power generation. In this work, we report the fabrication of hybrid systems based on chiral cyclo-dipeptide L-Tryptophan-L-Tryptophan incorporated into biopolymer electrospun fibers. The micro/nanofibers contain self-assembled nano-spheres embedded into the polymer matrix, are wide-band gap semiconductors with 4.0 eV band gap energy, and display blue photoluminescence as well as relevant piezoelectric and pyroelectric properties with coefficients as high as $57{\;\mathrm{CN}}^{-1}$ and $\;35\times 10^{-6}\;{\mathrm{Cm}}^{-2}K^{-1}$, respectively. Therefore, the fabricated hybrid mats are promising systems for future thermal sensing and energy harvesting applications.

## 1. Introduction

Nanotechnology is currently one of the most evolved scientific and technological fields in academic research and industry. Worldwide, the study of science and technology at the nanoscale has been one of the main areas in terms of research funding.

The use of peptides as a building block in nanotechnology has been increasingly exploited for its characteristics such as biocompatibility, flexibility and variability in molecular design [1,2,3].

Aromatic dipeptide chiral diphenylalanine (FF or PhePhe) and N-tert-butoxycarbonyl diphenylalanine (Boc-L-phenylalanine-L-phenylalanine-OH, Boc-PhePhe), have in their crystal structures phenylalanine molecules linked by directional covalent bonds spontaneously self-assembled into stable nanotubes (NT) and other nanostructures both in organic solvents and aqueous solutions [4,5,6,7].

The dipeptide assemblies form quantum confined (QC) structures with pronounced exciton effects because of directional intermolecular π-π interactions and a hydrogen-bonding network. These structures give rise to quantum dots (QD) through the formation of nanocrystalline regions with strong QC properties, resulting in a distinct blue luminescence. This phenomenon has been observed in dipeptides Boc-p-nitro-L-phenylalanyl-p-nitro-L-phenylalanine and Boc-L-phenylalanyl-L-tyrosine, which self-assemble into microspheres or microtapes [8].

Furthermore, hydrogen bonding networks in peptides and π-π interactions between aromatic moieties are responsible for the intrinsic semiconductor properties displayed by self-assembled dipeptides, polypeptides and proteins [9,10]. Consequently, dipeptide self-assemblies are direct-wide gap semiconductors, which form a new type of optoelectronic component and energy harvesting device. For example, phenylalanine-tryptophan (FW) nanostructures display higher conductivity than diphenylalanine (L-Phe-L-Phe, FF) due to their smaller band gap (3.04 eV), which is within the same range of energy as that of gallium nitride (3.39 eV) and silicon carbide (2.9 eV) [11]. Cyclic dipeptides (CDPs) are a group of ring-shaped dipeptides which, as a result of multiple interactions, can self-assemble with different functional architectures, such as nanospheres, nanoribbons, nanotubes and nanowires [12]. They exhibit semiconducting properties and improve stable photoluminescence (PL) in the visible region, relatively to their linear counterparts. These properties are attributed to the extensive hydrogen bonding and increased aromatic interactions that exist within their supramolecular structures. They are promising materials for optical wave guiding, nonlinear optics and biomechanical energy harvesting applications due to their high piezoelectric coefficients [11,13]. An example of this phenomenon is the cyclization of diphenylalanine. By introducing an aldehyde into self-assembling fibrous peptide networks, the result was the oriented crystallization into ordered superstructures that were uniaxially oriented along the longitudinal axis. This long-range oriented crystallization led to peptide platelets capable of functioning as optical waveguides as well as exhibiting blue photoluminescence emission under excitation at 330–380 nm [14].

More recently, biocompatible organic materials based on CDPs, self-assembled in supramolecular structures with different morphologies, showed very interesting quantum confinement and photoluminescence [12,15]. CDPs can self-assemble into various nanostructures by selecting specific amino acid side chains or through chemical modifications that provide several non-covalent interactions, such as hydrophobic interactions, π-π interactions and electrostatic forces [15,16]. The effect of external factors such as the pH, the substrate, the solvent and the temperature used during the process, among others, allow the formation of various nanostructures such as nanospheres (NS), nanotubes (NT) and nanofibers (NF) [17].

Nanostructures obtained from CDPs have a high potential for applications in several areas, such as energy storage devices, light emitting or display devices, use in hydrophobic surfaces for self-cleaning, piezoelectric devices, ultrasensitive sensors, in hydrogels for tissue engineering and drug delivery agents, among others [12,18]. Therefore, they are able to form functional architectures with wide applications in the field of materials science [19].

Dipeptide supramolecular structures that have in their constitution aromatic tryptophan amino acid, display enhanced properties such as high thermal stability and mechanical strength, conductivity and photoluminescence [2,3,20,21]. Tao and co-workers [2] showed that for cyclo-phenylalanine-tryptophan Cyclo (FW) and cyclo-tryptophan-tryptophan Cyclo (WW), had extensive and directional hydrogen bonds between backbone diketopiperazine rings together with aromatic side-chain interactions dimerized into quantum dots (QD) as a result of quantum confinement. Both dipeptides showed needle-like morphologies with a high aspect-ratio. However, of the two reported Cyclo (WW) dipeptides crystallized in a centrosymmetric space group ($P2_{1}/c$), only Cyclo (FW) crystallized in an acentric space group ($P2_{1})$. Consequently, only this last one exhibited the piezoelectric effect and had the characteristics to be configured as a peptide-based generator.

CDP scaffolds exhibit greater molecular rigidity in comparison to linear peptides, which can be attributed to the presence of four hydrogen bonding sites within the ring structure (comprising two donors and two acceptors) [17], as can be seen in Figure 1 for the molecular structure of chiral cyclic dipeptide cyclo-L-tryptophan-L-tryptophan formed by chiral tryptophan amino acids.

The simplest cyclic tryptophan-based dipeptide is cyclo-glycine-tryptophan Cyclo (GW), which forms crystals with a needlelike shape morphology from crystallization in a mixed methanol and water solution [3]. The crystals belong to the polar point group 2 and are thermally stable until 370 °C. They are photoluminescent with a maximum emission at 420 nm, in the blue light spectra region, when excited with wavelengths in the range of 300–400 nm. Due to its noncentrosymmetry, the crystal displays a high piezoelectric response, *d*_16_ and *d*_36_ approximately equal to 14 pC N^−1^ [3,11]. The aromatic packing networks present in Cyclo (GW) crystal supramolecular structure, promotes an output open-circuit voltage as high as 1.2 V under a force of 65 N applied periodically. Therefore, the crystals are suitable to be incorporated as active elements in energy harvester devices. 

In this work, we focused on the fabrication and characterization of hybrid systems based on nanostructured chiral cyclo-dipeptide cyclo-L-Tryptophan-L-Tryptophan incorporated into biopolymer fibers produced by the electrospinning technique. These nanofibers form anisotropic piezoelectric and pyroelectric self-assembled functional hybrid systems, suitable for thermal sensing and energy harvesting applications.

## 2. Experimental Section

### 2.1. Materials

Cyclic dioxopiperazine-L-tryptophan-L-tryptophan, herein referred to as Cyclo (L-Trp-L-Trp), was purchased from Bachem AG (Bubendorf, Switzerland). 1,4-dioxane was purchased from Fischer Chemicals (Zurich, Switzerland). *N,N*-dimethylformamide (DMF), dichloromethane (DCM), *N,N*-dimethylacetamide (DMAc), methanol and 1,1,1,3,3,3-hexafluoro-2-propanol (HFP) were purchased from Merck/Sigma-Aldrich (Darmstadt, Germany) and used as received.

Polycaprolactone (PCL, Mw 80,000) was purchased from Sigma-Aldrich. Poly-L-lactic acid (PLLA, Mw 217–225,000) was purchased from Corbion (Gorinchem, The Netherlands) and BDH Chemicals (Poole, UK), respectively.

### 2.2. Self-Assembling of Dipeptide Micro and Nanostructures in Solution

Fresh solution of Cyclo (L-Trp-L-Trp) was prepared by dissolving the dipeptide in HFP to a concentration of 100 mg/mL. This solution was then diluted in methanol (MeOH) to achieve the desired final concentrations based on the requirements of the studies to be conducted. The solutions were diluted in methanol and allowed to self-assemble for 24 h at room temperature. A few drops of a 5.37 mM solution of Cyclo (L-Trp-L-Trp) were placed on a silica slide and the solvent was removed by slow evaporation at room temperature and sent to SEM analysis and to Confocal Microscopy.

### 2.3. Electrospinning of Nanofibers

The 10% (*w*/*v*) polymer solution of PLLA was prepared by dissolving the polymer in a DCM/DMF solvent blend system (8:1, *v*/*v*) with vigorous stirring at room temperature. The 10% polymer solution of PCL was prepared by dissolving the polymer in a DMAc/DMF solvent blend system (7:1, *v*/*v*). After complete dissolution, Cyclo (L-Trp-L-Trp) dipeptide was incorporated into previously dissolved in DMF or DMAc in a 1:5 weight ratio. The incorporation of DMAc in some of the samples arises from the need to avoid precipitate formation in the solution during the addition of the dipeptide (some solutions became a milky color or a much higher viscosity than desirable). The precursor solutions obtained were stirred under ambient conditions for several hours prior to the electrospinning process.

The solutions obtained were loaded into a 5 mL syringe and connected to the anode of a high-voltage power supply (Spellmann CZE2000, Bochum, Germany) using a needle with an outer diameter of 0.5 mm and an inner diameter of 0.232 mm. The electrospinning apparatus had a vertical geometry, and the electrospinning process was performed at room temperature. To achieve stable spinning conditions and obtain fibers without beads, various parameters were adjusted, including the solution feeding flow rate, electric potential difference and needle-collector distance. An electric potential difference ranging from 18 to 20 kV, depending on the polymer and solvent ratio, was applied. The needle-collector distance was set at 11 cm and the flow rate was adjusted to between 0.15 and 0.40 mL/h. High-purity aluminum foil was attached to the collector to collect the prepared fibers, resulting in a random mesh of fibers on the foil. The aluminum foil also served as the electrodes during the electrospinning process, Figure 2.

### 2.4. Scanning Electron Microscopy (SEM)

The morphology, size, and shape of Cyclo (L-Trp-L-Trp) structures were studied using a Nova Nano SEM 200 scanning electron microscope operated at an accelerating voltage of 10 kV. Before SEM analysis, a thin film with a thickness of 10 nm consisting of Au-Pd alloy (80–20 weight%) was deposited onto the silica slide containing Cyclo (L-Trp-L-Trp) structures and the fiber mats using a high-resolution sputter coater (208HR Cressington Company) that was coupled to a Cressington MTM-20 high-resolution thickness controller. The diameter range of the Cyclo (L-Trp-L-Trp) structures and fibers was measured by SEM images using ImageJ 1.51n image analysis software (NIH, https://imagej.nih.gov/ij/, accessed on 19 October 2022). The average diameter and diameter distribution were determined by measuring 665 random nanospheres or 72–75 random fibers from the SEM images, and the results were fit to a log-normal function.

### 2.5. Mechanical Tests

A Zwick/Roell Z005 (ZwickRoell, Ulm, Germany) universal testing machine was used to access the mechanical properties of the electrospun fiber mat, following the ASTM D882-02 standard. Rectangular specimens (40 × 10) mm^2^ were cut from the fiber mat and tested under a crosshead velocity of 25 mm/min, with a gauge length of 26 mm. The results represent an average value of at least five specimens.

### 2.6. Optical Absorption and Photoluminescence

The Cyclo (L-Trp-L-Trp) solutions and fiber mats were subjected to optical absorption (OA) measurements using a Shimadzu UV-3101PC UV–Vis-NIR spectrophotometer (Shimadzu Corporation, Kyoto, Japan). Photoluminescence spectra were obtained using a Fluorolog 3 spectrofluorimeter (HORIBA Jobin Yvon IBH Ltd., Glasgow, UK).

For optical absorption measurements, Cyclo (L-Trp-L-Trp) solutions were prepared in methanol. The samples were measured in a quartz cuvette with 1 cm path length. Photoluminescence (PL) spectra acquired in the wavelength range of 290–600 nm using an excitation wavelength of 280 nm, with input and output slits fixed to provide a spectral resolution of 2 nm. Photoluminescence excitation (PLE) spectra were acquired in the wavelength range of 220–300 nm.

To measure the diffuse reflectance spectrum of the nanofiber mats in the wavelength range of 200–800 nm with a step size of 1 nm, a UV-2501PC spectrophotometer (Shimadzu Corporation, Kyoto, Japan) equipped with an integration sphere (Shimadzu ISR-205 240A) was used. Barium sulfate was used as a reference during the measurement. The energy of the band gap (E_g_) was calculated using the Kubelka-Munk function, which is expressed as [[hvF(R)]^1/2^] = α(hv − E_g_). Here, hv represents the energy of the incident photon, E_g_ corresponds to the energy of the band gap and F(R) is the Kubelka-Munk function. The Kubelka-Munk function was directly determined from the total reflectance coefficient of the material (R) using the equation F(R) = (1 − R)^2^/2R [22,23].

### 2.7. Confocal Laser Scanning Microscopy

The autofluorescence of the fibers was visualized using an Olympus FluoView FV1000 confocal scanning laser microscope (Olympus, Tokyo, Japan) with a 40× objective. The excitation wavelength was set at 405 nm and the detection filters were set to BA 430–470. The microscope captured images with a resolution of 800 × 800 pixels. A 1 cm^2^ fiber mat with a thickness of 600 μm was placed on a glass slide, and the sample was scanned at room temperature.

### 2.8. Dynamic Light Scattering (DLS) Measurements

The dipeptides were first prepared at a concentration of 3 µM in a solution of HFP/MeOH (0.2/9.8 *v*/*v*), and then 500 µL of this solution was diluted in water to a final concentration of 0.5 µM. The size, polydispersity and zeta potential of these solutions were then measured using a Litesizer 500 from Anton Paar (Baden-Wurttemberg, Germany). The equipment used a semiconductor laser diode with a wavelength of 658 nm and 40 mW, and a detection angle of 175°. The measurements were carried out at room temperature, and each sample was measured three times. The experimental data were processed using Kalliope™ software.

### 2.9. Pyroelectric Coefficient

Pyroelectricity is the property of some low-symmetry crystalline materials to convert a variation of temperature (T) within a certain time interval (dT/dt) into electric energy. The phenomena result from the temperature dependence of a material’s spontaneous polarization (P_s_). Changing the temperature results in compensation of the electric field that arises from intrinsic dipoles by the surface layer of free charges. The rate of change of the spontaneous polarization with temperature during heating or cooling, p = dP_s_/dT, is the pyroelectric coefficient. The change in polarization was detected by measuring the pyroelectric current I, which is proportional to the rate of change of polarization, using a Keithley 617 electrometer (Keithley Instruments GmbH, Landsberg, Germany). The equation used to calculate the pyroelectric current is I = A (dPs/dT)(dT/dt), where A is the electrode area and dT/dt is the rate of temperature change. The measurements were performed in a capacitor geometry under short-circuit conditions, meaning that the electrodes were connected with a wire of negligible resistance, which prevented any potential difference from arising across the sample during the measurement.

The fiber mat samples, with an area of (10 × 10) mm^2^ (200–330 µm thickness) formed a plane parallel capacitor covered with high purity metal plates.

### 2.10. Piezoelectric Output Voltage and Current

The piezoelectric output voltage and current were recorded using a low-pass filter and low-noise preamplifier (Research systems SR560, Stanford Research Systems, Stanford, CA, USA), and then captured by a digital storage oscilloscope (Agilent Technologies DS0-X-3012A, Waldbronn, Germany) after passing through a 100 MW load resistance. The sample of a fiber array had an area of 30 × 40 mm^2^ with a thickness ranging from 20 to 160 µm. It was subjected to periodic mechanical forces generated by a vibration generator (Frederiksen SF2185) with a frequency of 3 Hz, as determined by a signal generator (Hewlett Packard 33120A). Prior to the experiment, the applied forces were calibrated using a force-sensing resistor (FSR402, Interlink Electronics Sensor Technology, Graefelfing, Germany). During the electrospinning process, the fibers were deposited directly onto high-purity aluminum foil, which acted as the electrodes. The resulting samples were secured onto a stage, and uniform perpendicular forces were applied across the surface area of each sample. A piezoelectric nanogenerator, fabricated using an Cyclo (L-Trp-L-Trp)@PLLA electrospun fiber mat as the active piezoelectric component, is described in Figure 3. Thin high-purity copper plates were used as the top and bottom electrodes, with the top electrode having an area of 23 × 30 mm^2^ and the bottom electrode having an area of 30 × 33 mm^2^. Thin copper wires were then attached to these electrodes.

## 3. Results and Discussion

### 3.1. Morphology of Self-Assembled Dipeptide in Solution

The dipeptide cyclo-(L)-tryptophan-(L)-tryptophan, hereafter Cyclo (L-Trp-L-Trp) self-assembles as nanospheres (NS) with a 245 nm average diameter from a 5.4 mM methanol solution. Note that the nanospheres observed in the SEM image tend to self-organize and self-assembled into structures of larger dimensions, as can be seen in Figure 4a–c.

To better understand the self-assembling process of the Cyclo (L-Trp-L-Trp), the average size distribution of the dipeptide nanospheres was measured by dynamic light scattering (DLS). It is possible to make a comparison between the sizes of the nanospheres, obtained by SEM and by DLS; however, the latter only provides the hydrodynamic size, i.e., the size of the particles in movement. By utilizing this technique, it is possible to determine the size of particles dispersed in a liquid as a whole. To achieve this, the dipeptides were dissolved in HFP at room temperature. Methanol was then added to the solution to obtain a concentration of 3 µM, using a mixture of HFP and MeOH in a 0.2:9.8 *v*/*v* ratio. From this solution, 500 µL was diluted to a concentration of 0.5 µM in water, followed by ultrasonic treatment and DLS analysis [20]. The intensity-weighted distribution measurements indicated that the Cyclo (L-Trp-L-Trp) self-assembled as large molecular superstructures with several hundreds of nanometer size, have a mean hydrodynamic diameter of 283 nm, consistent with that obtained by SEM (245 nm) for dipeptide self-assembling as nanospheres (NS), Figure 5. Other parameters obtained were the transmittance of 75% and zeta potential of 19.6 ± 0.3 mV. A positive zeta potential demonstrates that Cyclo (L-Trp-L-Trp) NS possibly aggregate into a mass at the neutral condition. The obtained value was the desirable minimum zeta potential to achieve electrostatic and steric stabilization [24,25,26].

### 3.2. Morphology of Electrospun Fibers

During the electrospinning process, the flow of the polymer solution at the tip of the needle was steady, and there were no current fluctuations. The resulting fibers were white in appearance and exhibited a high degree of flexibility. Moreover, no beads or crystallites were observed on the surface of the fabricated fibers, Figure 6 and Figure 7.

Figure 7 displays scanning electron microscopy (SEM) images of fibers that were prepared using various polymers and embedded with Cyclo (L-Trp-L-Trp) dipeptide. The corresponding histograms of the diameter sizes are also presented alongside these images. The diameter distributions follow a log-normal dependence, with average values from 600 to 1014 nm. The fibers electrospun from the two polymers show, Figure 7d–f, average diameters from approximately 600 nm for Cyclo (L-Trp-L-Trp) embedded into PCL (Cyclo (L-Trp-L-Trp)@PCL), to thousands of nanometers (average 1014 nm) for cyclo-L-tryptophan-L-tryptophan embedded into PLLA (Cyclo (L-Trp-L-Trp)@PLLA), Figure 7a–c.

### 3.3. Mechanical Properties of Electrospun Fibers

The mechanical properties of the electrospun PCL nanofibers are shown in Figure 8, as an example. The stress-strain curves reveal a substantial improvement in mechanical performance (elasticity and ductility) of the nanofibers with the introduction of Cyclo (L-Trp-L-Trp) dipeptide inside the polymer matrix. The elastic modulus increased 188%, the tensile strength 221% and the elongation at break 42%. As previously shown in SEM analysis, the nanotube shape and good dispersion of cyclo-dipeptides explain the obtained mechanical results. The well-distributed cyclo-dipeptides NS with great interface are beneficial for the transfer of the applied forces between polymer and NS, having a reinforcing effect on the polymeric matrix. The present behaviour is similar to those from electrospun polymer fibers with embedded carbon nanotubes [27]. This variation is beneficial for the material piezoelectric response, allowing the fiber mat to withstand higher forces and deformations. It was found that the Young’s modulus depends significantly on the angle between the stretch direction and the fiber direction. In our case, the direction of stretching is the same as the fiber longitudinal axis orientation [28]. In our previous work, we have demonstrated that the incorporation of lead-free organic ferroelectric perovskite N-methyl-N′-diazabicyclo [2.2.2]octonium)-ammonium triiodide (MDABCO-NH_4_I_3_) nanocrystals embedded in polyvinyl chloride microfibers [29] and *N,N*-dimethyl-4-nitroaniline embedded in PLLA microfibers [30] improve the Young’s modulus compared to polymer microfibers alone. In this work we demonstrate that not only does the Young modulus increase, but all the mechanical properties measured for Cyclo (L-Trp-L-Trp)@PCL are improved as well.

### 3.4. Optical Absorption and Photoluminescence of Dipeptide Self-Assemblies

UV-Vis absorption spectra of Cyclo (L-Trp-L-Trp) nanospheres in solution at three different concentrations shows, over a broad maximum at 276 nm, three spike-like absorption peaks at 273 nm, 280 nm and 289 nm (indicating the formation of quantum dot (QD) structures, Figure 9a. As expected, the spike-like intensity increases with the increase of Cyclo (L-Trp-L-Trp) concentration in the methanol solution, indicating that the number of QD formed also increases, resulting in a higher contribution to the absorption spectra. Figure 9b shows the excitation spectra from a Cyclo (L-Trp-L-Trp) solution obtained for the maximum of emission at 312 nm reproducing the absorption spectra shown in Figure 9a. The observed spike-like peaks are consistent with those reported for Cyclo (WW) quantum confined structures reported in [20].

The dimensions of the quantum confined structures may be calculated for the present Cyclo (L-Trp-L-Trp) chiral dipeptide, from the correspondent OA and PLE spectra in Figure 9, following a theoretical model of quantum dots [31]. For Cyclo (L-Trp-L-Trp) QD, the calculated radius is R ≈ 1.41 nm, see SI. This is in agreement with those reported for QD formed in diphenylalanine (R ≈1.65 nm) [32] and for Cyclo (WW) (R ≈ 1.12 nm).

The band gap energy (Eg) may be calculated from the absorption spectra, through the Tauc plot [23], as shown in Figure 10 inset. Calculations indicate an energy of Eg = 4.068 ± 0.005 eV, slighlty higher than that calculated for Cyclo (FW) reported to be 3.1  eV using Density Functional Theory and reported to be 3.1  eV [2]. When Cyclo (L-Trp-L-Trp) is embedded into polymer fibers the optical band gap is aproximately 3.9 eV for both Cyclo (L-Trp-L-Trp)@PLLA and Cyclo (L-Trp-L-Trp)@PCL fibers, as shown in Figure 11a,b insets, obtained using the Kubelka-Munk function [22] (Appendix A). Therefore we may say that self-assembled nanostructures of Cyclo (L-Trp-L-Trp) are bioorganic wide-band gap semiconductors [33,34].

The QD intrinsic photoluminescence is shown in Figure 12, where the emission spectra were measured for four different dipeptide concentrations, for excitation at 280 nm. Besides a maximum at 310 nm, there are also two less intense broad peaks at around 323 nm, and 340 nm resulting from the QD emitting structures. The intensities also increase with the increasing concentration as a result of an increase of the number of QC structures in the solution. These photoluminescent properties results from the organized QD inside the self-assembled structures. The confocal microscopy image, Figure 12b, shows that Cyclo (L-Trp-L-Trp) NS formed from the self-assembling of QD resulting from the dipeptide dimerization.

Aiming at understanding the dipeptide self-assembly when embedded into the polymer fiber matrix, OA and PL spectra of Cyclo (L-Trp-L-Trp) embedded into PCL fibers was measured after polymer dissolution in DCM/MeOH (4:1 *v*/*v*), as a function of time. Similar to Figure 9a, a broad band with maximum at 276 nm contains three spike-like absorption peaks at 273 nm, 279 nm and 289 nm indicating the formation of QD structures, Figure 13a. The number of QD formed also increases with time resulting in an increasing of intensity over time. PL spectrum (under excitation at 280 nm) displayed by the self-assembled nanostructures formed inside the Cyclo (L-Trp-L-Trp) nanofibers is presented in Figure 13b. Here an emission band with a maximum at 340 nm increases in intensity with time, accompanied by a 30 nm shift relative to the PL maximum from the cyclo dipeptide self-assembly in solution (which is 310 nm). Interestingly, there is also an evolution of PL emission after times over 18 min where a broader small PL band grows at the expense of the previous one and shifts to higher wavelengths, with a maximum at 402 nm with time. This broader band results from crystallization of the dipeptide nanostructures inside the fibers. Fluorescence confocal microscopy, Figure 13c, indeed shows blue luminescence from the dipeptide nanostructures crystallized inside the fibers.

### 3.5. Pyroelectricity in Fibers

In this work, we report for the first time the measurement of the pyroelectric effect of Cyclo (L-Trp-L-Trp) dipeptide embedded in PLLA and PCL fibers. The measured coefficients, as a function of temperature, are shown in Figure 14a,b for Cyclo (L-Trp-L-Trp)@PCL and Cyclo (L-Trp-L-Trp)@PLLA, respectively. Although the crystal structure of Cyclo (L-Trp-L-Trp) is not known, the existence of a pyroelectric effect indicates that the crystal point group is one of the eleven polar crystallographic point groups and therefore the dipeptide might also display ferroelectricity.

The pyroelectric coefficient of the Cyclo (L-Trp-L-Trp)@PCL and Cyclo (L-Trp-L-Trp)@PLLA reaches 35 × 10^−6^ Cm^−2^k^−1^ at 323 K and 36 × 10^−6^ Cm^−2^k^−1^ at 315 K, respectively, which is far below the melting temperature of the cyclic dipeptide. These pyroelectric coefficient values are roughly four times higher than that reported for electrospun poly(vinyl alcohol) (PVA) nanofibers doped with the hydrogen bonded molecular ferroelectric 1,4-diazabicyclo [2.2.2]-octane perrhenate (dabcoHReO_4_), which is 8.5 × 10^−6^ Cm^−2^k^−1^ at 300 K [35]. Remarkably, the pyroelectric coefficient of polycrystalline dipeptide Cyclo (L-Trp-L-Trp) embedded into polymer fibers is only one order of magnitude smaller than that reported for the state-of-the-art semiorganic ferroelectric triglycine sulfate (TGS) oriented single crystal, reported to be 306 × 10^−6^ Cm^−2^k^−1^ at the ferroelectric-paraelectric phase transition (322 K) [36].

For characterization of pyroelectric materials into thermal sensor devices, a figure-of-merit (FOM), defined as $FOM=p/\sqrt{\epsilon_{r}}$ where p is the pyroelectric coefficient and $\epsilon_{r}$ is the relative dielectric constant, can be calculated. Accordingly, substituting the value of $p=35\times 10^{-6\;}{\mathrm{Cm}}^{-2}K^{-1}$ and assuming a relative dielectric constant of 10 for the dipeptide, we obtain $\;FOM=11\times 10^{-6\;}{\mathrm{Cm}}^{-2}K^{-1}$ for Cyclo (L-Trp-L-Trp)@PLLA and Cyclo (L-Trp-L-Trp)@PCL electrospun fiber mats. This value is comparable to that reported for PVDF and P(VDF-TrFE) 50/50 polymer at $33.24\times 10^{-6\;}{\mathrm{Cm}}^{-2}K^{-1}$ and $5.06\times 10^{-6\;}{\mathrm{Cm}}^{-2}K^{-1}$, respectively [37,38].

Importantly, the pyroelectric coefficient measured on a bundle of diphenylalanine microtubes was reported to be around $2\times 10^{-6\;}{\mathrm{Cm}}^{-2}K^{-1\;}$ [39], one order of magnitude smaller than that presented in this work for Cyclo (L-Trp-L-Trp)@PLLA and Cyclo (L-Trp-L-Trp)@PCL electropsun fibers. Our results are significant and indicate that the electrospun fiber mats are potential candidates to integrate into pyroelectric sensing [40]. Moreover, this is the first work reporting the pyroelectric properties of a cyclo-dipeptide.

### 3.6. Piezoelectric Voltage and Effective Piezoelectric Coefficients in Fibers

The electrospun Cyclo (L-Trp-L-Trp)@PLLA and Cyclo (L-Trp-L-Trp)@PCL nanofiber mats, fabricated with the active piezoelectric dipeptide, are now demonstrated to behave as piezoelectric energy generators transforming mechanical energy, due to a periodically applied force per unit area, into electric energy. When the fiber mats are compressed or released, there is a reorientation of the molecular dipoles within the crystalline Cyclo (L-Trp-L-Trp) dipeptide material followed by a charge separation which originates from an output voltage and, consequently, an external electric current outside is generated through an external circuit. In the present work, we demonstrate that incorporating Cyclo (L-Trp-L-Trp) nanostructures into electrospun polymer fibers, processed at room temperature without poling, is an easy and straightforward way to fabricate piezoelectric generators using organic biomolecules as active materials. Figure 15 shows that for Cyclo (L-Trp-L-Trp)@PLLA and Cyclo (L-Trp-L-Trp)@PCL nanofiber mats, the maximum open-circuit voltage and current, measured through a load resistance of 100 MΩ reaches 11.5V and 115 nA, and 9.6 V and 96 nA under periodically compressive forces of 2 N (5 kPa) and 3.2 N (8 kPa), respectively. PCL is not a piezoelectric polymer and therefore all the piezoelectric voltage generated by Cyclo (L-Trp-L-Trp)@PCL fibers is due to the dipeptide. PLLA is itself a piezoelectric polymer having a small contribution to the piezoelectric response in the case of Cyclo (L-Trp-L-Trp)@PLLA fibers. Figure 15c shows that the output voltage is proportional to the applied forces, thus confirming the linear piezoelectric properties of Cyclo (L-Trp-L-Trp). We can calculate the piezoelectric charge, with the equation Q = ∫Idt (C), generated by the piezoelectric mats upon compression and using the maximum intensity achieved from Figure 15a,b during a 10^−3^ s material response time. For Cyclo (L-Trp-L-Trp)@PLLA and Cyclo (L-Trp-L-Trp)@PCL fibers, the charges generated are 115 pC and 96 pC when the applied force are 3.2 N and 2.0 N, respectively. Therefore, the correspondent effective or average piezoelectric coefficients, defined as d_eff_ = Q/F (pCN^−1^), are 57 pCN^−1^ and 30 pCN^−1^, as indicated in Table 1. These coefficients are higher than those predicted for dipeptides Cyclo (GW) oriented single crystals (on average 5.6 pCN^−1^) and measured in biological materials such as viruses (8 pCN^−1^) and rat tail collagen (12 pCN^−1^ [3]). Furthermore, they are also comparable to those reported for lead-free organic ferroelectric perovskite N-methyl-N′-diazabicyclo [2.2.2]octonium)-ammonium triiodide (MDABCO-NH_4_I_3_) nanocrystals embedded in polyvinyl chloride (MDABCO-NH_4_I_3_@PVC), which is 64 pCN^−1^. Another quantity that is interesting to calculate is the peak power density given by $P=\left(RI^{2}\right)/A\;$(μWcm^−2^), where R = 100 MΩ is the load resistance and A is the electrode’s area, delivered by the nanofiber mats. These values are 0.18 μWcm^−2^ and 0.13 μWcm^−2^ for Cyclo (L-Trp-L-Trp)@PLLA and Cyclo (L-Trp-L-Trp)@PCL fibers, respectively.

Finally, and very importantly, is the piezoelectric voltage coefficient g_eff_ = d_eff_/(ε’ε_0_) VmN^−1^, a figure of merit which quantifies the materials’ adequacy as a piezoelectric sensor. Assuming a dielectric constant of ε´~ 10 at 10 Hz (common to dipeptides in Table 1) we obtain g_eff_ = 4.7 VmN^−1^ and g_eff_ = 2.6 VmN^−1^ for Cyclo (L-Trp-L-Trp)@PLLA and Cyclo (L-Trp-L-Trp)@PCL fiber mats, respectively. These values are within the same order of magnitude of those obtained for the crystal powder Boc-Dip-Dip nanogenerator and higher than that for the polycrystalline Cyclo (GW) nanogenerator, as shown in Table 1. One should notice that, remarkably, our results were measured using compressive forces per unit area two orders of magnitude smaller than those used for practical applications; an important parameter to consider when using organic crystals therefore avoiding their damage over time. In fact, as an example, Appendix A shows that the Cyclo (L-Trp-L-Trp)@PLLA fiber mat has an unchangeable, continuous piezoelectric output voltage over a long period of time. We may conclude that cyclic dipeptide Cyclo (L-Trp-L-Trp) nanocrystals embedded into biopolymers and fabricated by the electrospinning technique, form easily handled nanomaterials systems capable of performing as piezoelectric energy nanogenerators with relevant piezoelectric voltage coefficients.

## 4. Conclusions

In summary, we demonstrate that chiral cyclo (L-Tryptophan-L-Tryptophan) dipeptide when embedded into PLLA and PCL biopolymer electrospun fibers, namely Cyclo (L-Trp-L-Trp)@PLLA and Cyclo (L-Trp-L-Trp)@PCL, form very flexible mats with improved mechanical stability. The chiral cyclic dipeptide self-assemblies, both in solution and when embedded into the polymer matrix as nanospheres, with a mean hydrodynamic diameter of 283 nm, display quantum confinement. The nanospheres show blue luminescence in the solid state. The dimensions of the quantum-confined structures calculated from the corresponding optical spectra, indicate that the cyclo-dipeptide quantum dots have a radius approximately equal to 1.41 nm, in agreement with the values reported for dipeptide diphenylalanine nanotubes (1.65 nm) and centrosymmetric cyclo-tryptophan-tryptophan needle-shaped crystals (1.12 nm).

The band gap energy of cyclo (L-Tryptophan- L-Tryptophan) nanospheres, both free and when embedded into the fibers, is around 4 eV, close to that calculated for cyclo (L-phenylalanine-tryptophan) which is 3.1 eV. Therefore, self-assembled nanostructures of Cyclo (L-Trp-L-Trp) are bioorganic wide-band gap semiconductors.

In this work we demonstrated that the electrospun fabricated nano/microfiber Cyclo (L-Trp-L-Trp)@PLLA and Cyclo (L-Trp-L-Trp)@PCL mats, are potential energy harvesting systems both as piezoelectric and pyroelectric active hybrid functional materials. Their piezoelectric functionality, accessed under the application of periodical forces of several Newtons, indicated a strong response resulting from measured average piezoelectric coefficients as high as 57 pCN^−1^ and 30 pCN^−1^. These coefficients are comparable to those reported for lead-free organic ferroelectric perovskite *N*-methyl-*N*′-diazabicyclo [2.2.2]octonium)-ammonium triiodide (MDABCO-NH_4_I_3_) nanocrystals embedded in polyvinyl chloride (MDABCO-NH_4_I_3_@PVC), which is 64 pCN^−1^. Moreover, the piezoelectric voltage coefficient, a figure of merit which quantifies the materials adequacy as a piezoelectric sensor is *g_eff_* = 4.7 VmN^−1^ and *g_eff_* = 2.6 VmN^−1^ for Cyclo (L-Trp-L-Trp)@PLLA and Cyclo (L-Trp-L-Trp)@PCL fiber mats, respectively. These values are higher than that obtained for a polycrystalline Cyclo (GW) nanogenerator of 1.6 VmN^−1^.

For potential applications as integrated micro bioorganic thermal sensors, a high pyroelectric coefficient and thermal stability are of high importance. Measurements of the pyroelectric current on the fabricated electrospun hybrid fiber mats, revealed that the pyroelectric coefficient reaches $35\times 10^{-6}{\;\mathrm{Cm}}^{-2}K^{-1}\;$ and $36\times 10^{-6}{\;\mathrm{Cm}}^{-2}K^{-1}$, for Cyclo (L-Trp-L-Trp)@PCL and Cyclo (L-Trp-L-Trp)@PLLA, respectively, far below the melting temperature of the cyclo-dipeptide. Remarkably, these values are one order of magnitude larger than that reported for a bundle of diphenylalanine microtubes which was $2\times 10^{-6}\;{\mathrm{Cm}}^{-2}K^{-1}$. Moreover, the present work reports for the first time the pyroelectric properties of a polar cyclo-dipeptide made from two chiral l-tryptophan amino acids.

Chiral cyclo-L-Tryptophan- L-Tryptophan dipeptide nanocrystals, when embedded into biopolymers using the electrospinning technique, form stable and easily handled nano/micro hybrid all-organic biomaterial systems capable of future incorporation into piezoelectric energy harvesting and pyroelectric temperature sensing devices.

## Figures and Tables

**Figure 1 materials-16-02477-f001:**
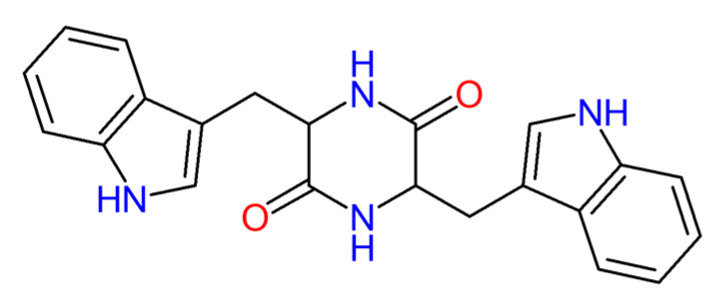
Chemical structure of the chiral dipeptide cyclo-L-Tryptophan-L-Tryptophan.

**Figure 2 materials-16-02477-f002:**
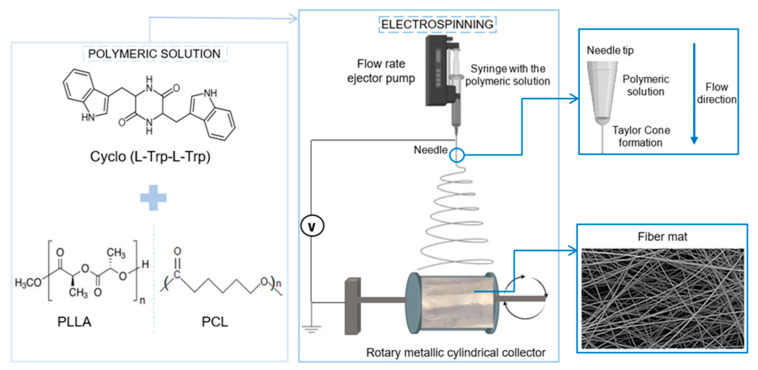
Schematic flow chart for the preparation of Cyclo (L-Trp-L-Trp))@PLLA Cyclo (L-Trp-L-Trp)@PCL electrospun nanofibers.

**Figure 3 materials-16-02477-f003:**
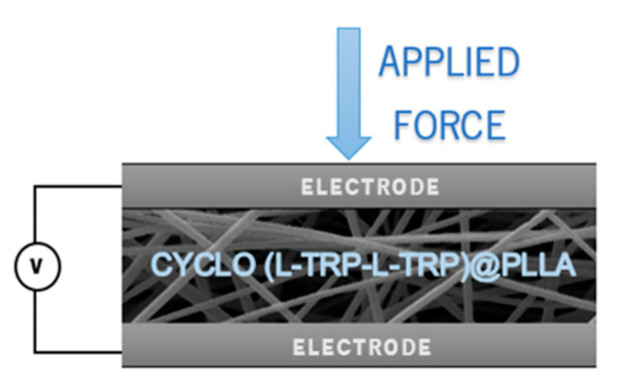
Schematic of Cyclo (L-Trp-L-Trp)@PLLA piezoelectric nanogenerator. Electrospun nanofiber mat sandwiched between two copper electrodes.

**Figure 4 materials-16-02477-f004:**
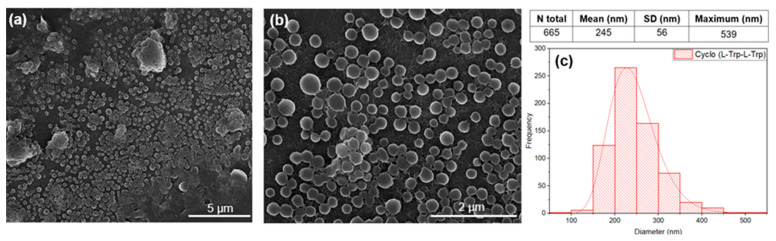
SEM images at magnification levels of 15,000× (**a**) and 50,000× (**b**), and the diameter distribution (**c**) of the nanospheres (NS) of Cyclo (L-Trp-L-Trp), self-assembled in methanol solution. The logarithmic normal distribution is indicated by the red curve.

**Figure 5 materials-16-02477-f005:**
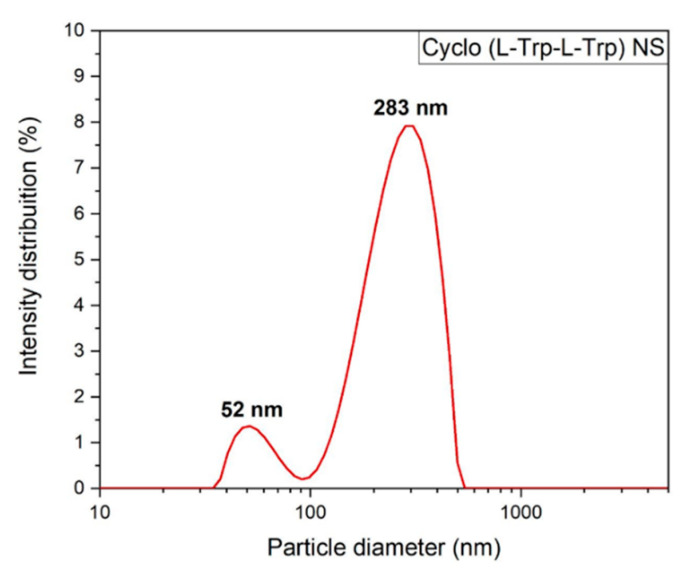
Intensity weighted particle size distributions for Cyclo (L-Trp-L-Trp) nanospherical structures measured by dynamic light scattering.

**Figure 6 materials-16-02477-f006:**
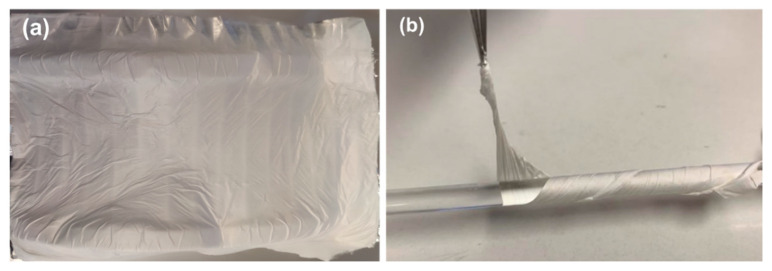
Cyclo (L-Trp-L-Trp)@PLLA electrospun fiber mat (**a**) and Cyclo (L-Trp-L-Trp)@PCL fiber mat folded around a cylindrical rod, showcasing the flexibility of the fibers (**b**).

**Figure 7 materials-16-02477-f007:**
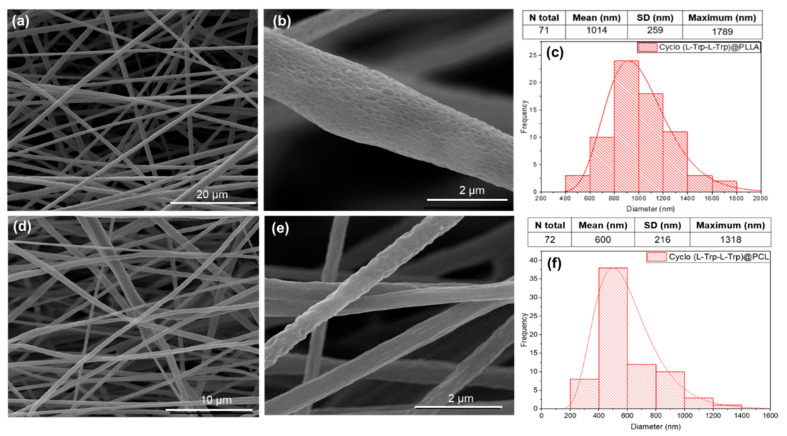
SEM images at magnification levels of 5000× (**a**), 10,000× (**d**) and 50,000× (**b**,**e**) and the respective fiber diameter distribution histograms for PLLA (**a**–**c**) and PCL (**d**–**f**) with embedded Cyclo (L-Trp-L-Trp) dipeptide. The logarithmic normal distributions for each set of fibers are represented by the red curves, which were generated using the mean, maximum and standard deviations of the data.

**Figure 8 materials-16-02477-f008:**
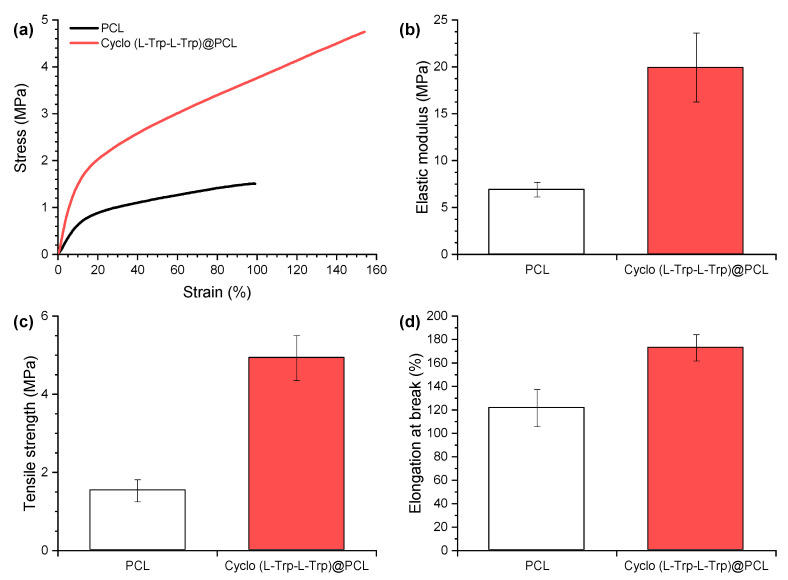
Mechanical properties of electrospun fibers of PCL (in black) and Cyclo (L-Trp-L-Trp)@PCL (in red): (**a**) representative tensile curves, (**b**) elastic modulus, (**c**) tensile strength and (**d**) elongation at break.

**Figure 9 materials-16-02477-f009:**
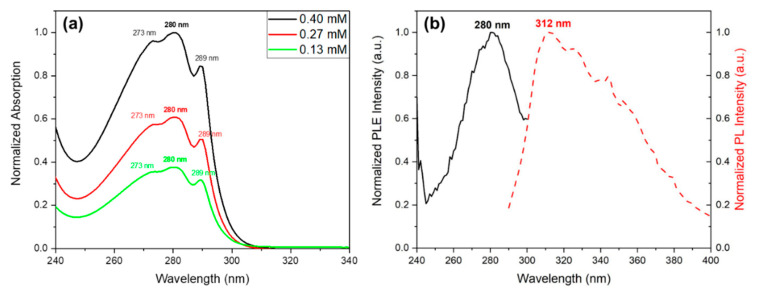
Cyclo (L-Trp-L-Trp) dipeptide in MeOH (**a**) normalized UV-vis absorption spectra at different solution concentrations; (**b**) normalized excitation and emission spectra. The emission wavelength is 312 nm.

**Figure 10 materials-16-02477-f010:**
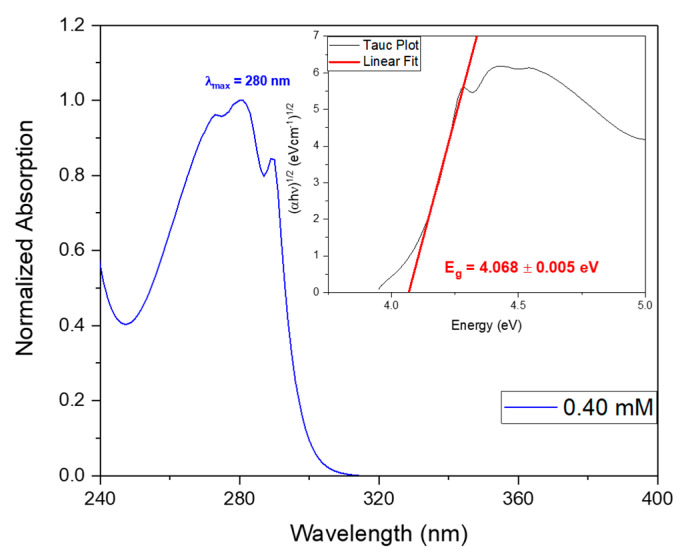
Normalized UV-vis absorption spectrum of Cyclo (L-Trp-L-Trp) dipeptide at the concentration of 0.40 mM. The inset shows the band gap energy calculated from the Tauc plot.

**Figure 11 materials-16-02477-f011:**
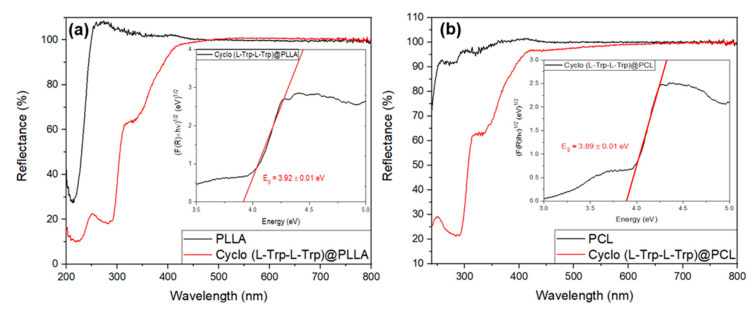
Reflectance spectra of (**a**) Cyclo (L-Trp-L-Trp)@PLLA and (**b**) Cyclo (L-Trp-L-Trp)@PCL fibers. The inset shows the band gap energy calculated from the Kubelka-Munk function with the linear fit in red.

**Figure 12 materials-16-02477-f012:**
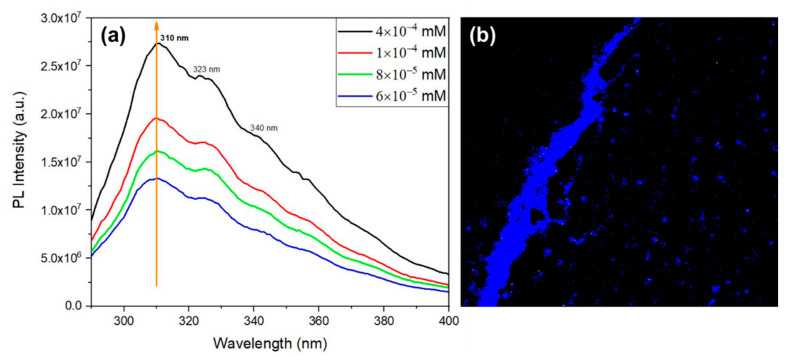
(**a**) Photoluminescent emission spectra of Cyclo (L-Trp-L-Trp) in MeOH at several concentrations. The excitation wavelength is 280 nm. (**b**) Confocal microscopy image of Cyclo (L-Trp-L-Trp) crystallized nanospheres, under laser excitation at 405 nm.

**Figure 13 materials-16-02477-f013:**
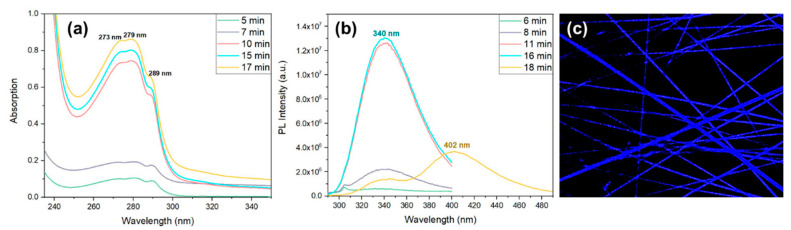
(**a**) Optical Absorption and (**b**) photoluminescence spectra as a function of time, from Cyclo (L-Trp-L-Trp) embedded into PCL fibers, after polymer fiber dissolution in DCM/MeOH (4:1 *v*/*v*). The excitation wavelength is 280 nm. (**c**) Confocal microscopy image of Cyclo (L-Trp-L-Trp)@PCL fiber mats, under laser excitation at 405 nm.

**Figure 14 materials-16-02477-f014:**
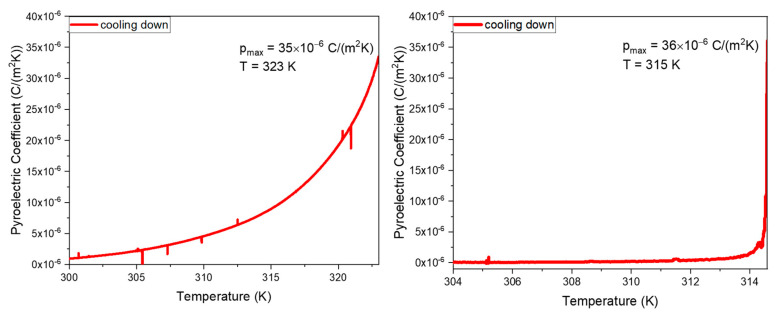
Pyroelectric coefficient as function of temperature (measured on cooling) of (**a**) Cyclo (L-Trp-L-Trp)@PCL and (**b**) Cyclo (L-Trp-L-Trp)@PLLA.

**Figure 15 materials-16-02477-f015:**
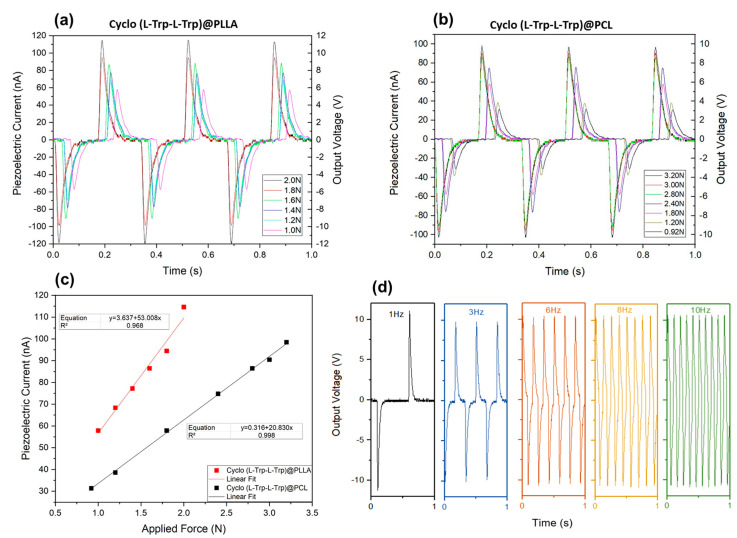
(**a**,**b**) Piezoelectric current and output voltage as a function of time; (**c**) piezoelectric current as a function of different applied periodical forces from Cyclo (L-Trp-L-Trp)@PLLA and Cyclo (L-Trp-L-Trp)@PCL; (**d**) output voltage for low frequencies up to 10 Hz from Cyclo (L-Trp-L-Trp) incorporated into electrospun PLLA polymer fibers.

**Table 1 materials-16-02477-t001:** Piezoelectric nanogenerator parameters for some dipeptides.

Nanogenerator	Force/Area (N/m^2^)	V_out_ (V)	*d*_eff_ (pC/N)	*g*_eff_ (Vm/N)	Power Density (μWcm^−2^)	Ref.
Cyclo (L-Trp-L-Trp)@PLLA (fiber mat)	3 × 10^3^	11.5	57	4.7	0.18	This work
Cyclo (L-Trp-L-Trp)@PCL (fiber mat)	5 × 10^3^	9.6	30	2.6	0.13	This work
Cyclo (GW) (crystal powder)	7 × 10^5^	1.2	5.6 ^§^	1.6 ^#^	0.002	[3]
Cyclo (FW) (crystal powder)	6 × 10^5^	1.4	16 *	1.3 **	0.003	[2]
Boc-PhePhe@PLLA (fiber mat)	4 × 10^3^	30	8.4	0.3	2.3	[8]
Boc-PheTyr@PLLA (fiber mat)	4 × 10^3^	24	7	0.3	1.0	[8]
Boc-*p*NPhe*p*NPhe@PLLA (fiber mat)	4 × 10^3^	58	16	0.6	9.0	[8]
Boc-DipDip (crystal powder)	4 × 10^4^	1	73	2.8	__	[41]
MDABCO-NH_4_I_3_@PVC (fiber mat)	11 × 10^3^	16.5	175	3.6	0.20	[29]

* Calculated from data available in [2], assuming a nanogenerator time response of 0.5 s; ** calculated from data available in [2], assuming a dielectric constant around 10; ^§^ calculated average piezoelectric coefficient from [3]; ^#^ calculated assuming a dielectric constant of 2.9 from [3].

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
