# Peer review of "Bioinspired Cyclic Dipeptide Functionalized Nanofibers for Thermal Sensing and Energy Harvesting"

_materials, 2023, doi:10.3390/ma16062477_

Round 1

Reviewer 1 Report

Reviewer Comments to Author

The topic of the paper is interesting.  In general, the main concept of the work is interesting but Authors should improved some elements – all suggestions are given below in more detail:

1)             Abstract of the paper should be supplemented with the novelty of the topic.

2)             References should contain more open access articles, e.g. from journals such as Materials, but not only from MDPI.

3)             Conclusions should be more quantified.

4)             Authors should make some editorial corrections.

Author Response

Response to Reviewer 1

We thank you very much for your comments and suggestions, which we address below.

"The topic of the paper is interesting. In general, the main concept of the work is interesting but Authors should improve some elements – all suggestions are given below in more detail:

1) Abstract of the paper should be supplemented with the novelty of the topic. Authors Response: The following sentences were added to the abstract:
Cyclic dipeptides are an emerging outstanding group of ring-shaped dipeptides, which because of multiple interactions; self-assemble in supramolecular structures with different morphologies, showing quantum confinement, photoluminescence. Chiral cyclic dipeptides may also display piezoelectricity and pyroelectricity, properties with potential applications in new sources of nanoenergy. Modifications are highlighted in yellow in the revised manuscript.

2) References should contain more open access articles, e.g. from journals such as Materials, but not only from MDPI. Authors Response: More open access references are added and highlighted in yellow: references [16], [18] and [21]. Modifications are highlighted in yellow in the revised manuscript.

3) Conclusions should be more quantified. Authors Response: The conclusions have been rewritten in a more quantified and clear way putting in evidence the novelty and potential applications of the present study. Modifications are highlighted in yellow in the revised manuscript.

4) Authors should make some editorial corrections. Authors Response: Editorial corrections were made throughout the all manuscript and are highlighted in yellow in the revised manuscript.

Reviewer 2 Report

In the manuscript entitled “Bioinspired Cyclic dipeptide functionalized nanofibers for thermal sensing and energy harvesting”, the authors studied how self-assembled nanostructures based on chiral cyclic dipeptides (CDPs) can be used as optoelectronic materials for energy harvesting devices. The authors synthesized hybrid systems based on L-Tryptophan-L-Tryptophan CDPs incorporated into biopolymer electrospun fibers. The micro/nanofibers were characterized by various techniques, and found that their fibers contain self-assembled nano-spheres embedded into the polymer matrix that are wide-bandgap semiconductors with blue photoluminescence emission, reporting that their fibers have high piezoelectric and pyroelectric coefficients that make them suitable for thermal sensing and energy harvesting applications. I can recommend the publication of the manuscript on Materials providing the following comments and questions are addressed:

1. Please note the grammatical errors, e.g. "They are therefore promise systems for thermal sensing and energy harvesting applications." in the abstract. "Please try to ensure the correctness of the abstract and conclusion.

2. Please uniform drawing format, and improve the clarity of the pictures, some of the graphs are wrongly labeled and the legend information is unclear.

3. It is well known that the response time of a sensor is one of its important parameters, please analyze the response time of the material as a prepared thermal sensor device.

4. Please explain why the material is considered a semiconductor material with indirect leap when using the Tauc plot formula. If you have a reference, please indicate the source.

5. Please concise description of experimental results, with an appropriate analysis of picture results.

6. Why do the peaks of piezoelectric current and output voltage in Figure 15a, b appear to be shifted, and what is the reason for this?

Author Response

Response to Reviewer 2

We thank you very much for your comments and suggestions, which we address below.

“In the manuscript entitled “Bioinspired Cyclic dipeptide functionalized nanofibers for thermal sensing and energy harvesting”, the authors studied how self-assembled nanostructures based on chiral cyclic dipeptides (CDPs) can be used as optoelectronic materials for energy harvesting devices. The authors synthesized hybrid systems based on L-Tryptophan-L-Tryptophan CDPs incorporated into biopolymer electrospun fibers. The micro/nanofibers were characterized by various techniques, and found that their fibers contain self-assembled nano-spheres embedded into the polymer matrix that are wide-bandgap semiconductors with blue photoluminescence emission, reporting that their fibers have high piezoelectric and pyroelectric coefficients that make them suitable for thermal sensing and energy harvesting applications. I can recommend the publication of the manuscript on Materials providing the following comments and questions are addressed:

  1. Please note the grammatical errors, e.g. "They are therefore promise systems for thermal sensing and energy harvesting applications." in the abstract. "Please try to ensure the correctness of the abstract and conclusion.

Authors Response: The correctness of the abstract and conclusions have been addressed. Modifications are highlighted in yellow in the revised manuscript.

  1. Please uniform drawing format, and improve the clarity of the pictures, some of the graphs are wrongly labelled and the legend information is unclear.

Authors Response: As requested all the drawings were formatted, the clarity of the figures improved and corrections made thoroughly the manuscript.

  1. It is well known that the response time of a sensor is one of its important parameters, please analyze the response time of the material as a prepared thermal sensor device.

Authors Response: A paragraph with the calculated FOM for the electrospun fibers was introduced and reads as:

“For characterization of pyroelectric materials into thermal sensor devices, a figure-of-merit () defined as  where  is the pyroelectric coefficient and  the relative dielectric constant, can be calculated. Accordingly substituting the values of  and assuming a relative dielectric constant of 10 for the dipeptide, we obtain for Cyclo (L-Trp-L-Trp)@PLLA and Cyclo (L-Trp-L-Trp)@PCL electrospun fiber mats. This value is comparable to that reported for PVDF as and P(VDF-TrFE) 50/50 terpolymer, respectively and

In the present work, we have not yet realized a prototype device for thermal sensing. This will be the next step to be carried on in the very near future, where a detailed study will made for practical applications.

  1. Please explain why the material is considered a semiconductor material with indirect leap when using the Tauc plot formula. If you have a reference, please indicate the source.

Authors Response: A reference has been added related to the calculation of the energy band gap from the absorption spectra. Ref [23]. https://doi.org/10.1021/acs.jpclett.8b02892.

  1. Please concise description of experimental results, with an appropriate analysis of picture results.

Authors Response: We tried to address this point as best as we could. However, we think that the figure captions should be looked in a dynamic way by interchanging it with the text.

  1. Why do the peaks of piezoelectric current and output voltage in Figure 15a, b appear to be shifted, and what is the reason for this?

Authors Response: The peaks in Figure 15 a) and b) have been shifted on purpose for a clearer demonstration that their maximum values depend on the magnitude of the applied forces.

Round 2

Reviewer 1 Report

Manuscript has been corrected according to the recommendations of the reviewer. All suggestions of the reviewer have been analyzed. In conclusion, revised version of the manuscript may be accepted for publication in the journal.

Reviewer 2 Report

No further comment.